# Chromatin Interaction Changes during the iPSC-NPC Model to Facilitate the Study of Biologically Significant Genes Involved in Differentiation

**DOI:** 10.3390/genes11101176

**Published:** 2020-10-08

**Authors:** Won-Young Choi, Ji-Hyun Hwang, Jin-Young Lee, Ann-Na Cho, Andrew J Lee, Inkyung Jung, Seung-Woo Cho, Lark Kyun Kim, Young-Joon Kim

**Affiliations:** 1Interdisciplinary Program of Integrated OMICS for Biomedical Science, The Graduate School, Yonsei University, Seoul 03722, Korea; cyberon21@gmail.com (W.-Y.C.); jiihyun@gmail.com (J.-H.H.); 2Department of Biochemistry, College of Life Science and Biotechnology, Yonsei University, Seoul 03722, Korea; hd00ljy@naver.com; 3Department of Biotechnology, College of Life Science and Biotechnology, Yonsei University, Seoul 03722, Korea; annnacho105@gmail.com (A.-N.C.); seungwoocho@yonsei.ac.kr (S.-W.C.); 4Department of Biological Sciences, KAIST, Daejeon 34141, Korea; andrewjlee1213@gmail.com (A.J.L.); ijung@kaist.ac.kr (I.J.); 5Severance Biomedical Science Institute and BK21 PLUS Project for Medical Sciences, Gangnam Severance Hospital, Yonsei University College of Medicine, Seoul 06230, Korea

**Keywords:** Hi-C, spatial organization, chromatin interactions, human induced pluripotent stem cells, neural progenitor cells, neuronal differentiation

## Abstract

Given the difficulties of obtaining diseased cells, differentiation of neurons from patient-specific human induced pluripotent stem cells (iPSCs) with neural progenitor cells (NPCs) as intermediate precursors is of great interest. While cellular and transcriptomic changes during the differentiation process have been tracked, little attention has been given to examining spatial re-organization, which has been revealed to control gene regulation in various cells. To address the regulatory mechanism by 3D chromatin structure during neuronal differentiation, we examined the changes that take place during differentiation process using two cell types that are highly valued in the study of neurodegenerative disease - iPSCs and NPCs. In our study, we used Hi-C, a derivative of chromosome conformation capture that enables unbiased, genome-wide analysis of interaction frequencies in chromatin. We showed that while topologically associated domains remained mostly the same during differentiation, the presence of differential interacting regions in both cell types suggested that spatial organization affects gene regulation of both pluripotency maintenance and neuroectodermal differentiation. Moreover, closer analysis of promoter–promoter pairs suggested that cell fate specification is under the control of cis-regulatory elements. Our results are thus a resourceful addition in benchmarking differentiation protocols and also provide a greater appreciation of NPCs, the common precursors from which required neurons for applications in neurodegenerative diseases such as Parkinson’s disease, Alzheimer’s disease, schizophrenia and spinal cord injuries are utilized.

## 1. Introduction

The use of patient-specific human induced pluripotent stem cells (iPSCs) and their differentiated derivatives in disease-modelling and cell therapy development has been gaining traction in the study of neurodegenerative diseases as the cells involved are difficult to isolate and scarce in number [1]. Of special interest are neural progenitor cells (NPCs)—precursor cells from which neurons are generated for applications in diseases such as Parkinson’s disease [2], Alzheimer’s disease [3,4], schizophrenia [5] and spinal injury [6]. Comprehensive understanding of the iPSC-NPC model is thus crucial and other groups have tracked changes during the differentiation process through cellular [7] and transcriptomic studies [8]. However, there are limitations in making nuanced interpretations of interactions between cis-regulatory elements and various studies have shown that gene regulation is closely related to genome spatial organization [9,10,11]. Genome-wide 3D chromosomal construct mapping technologies have unearthed crucial findings on chromosome folding [12,13], and it has been reported that spatial conformation of chromatin is highly related with gene regulation such as promoter–promoter interaction and promoter–enhancer interaction [14,15]. During stem cell differentiation, chromatin interaction changes can be observed in defined regions called topologically associated domains (TAD) and these changes affect the activity of genes related in differentially interacted regions (DIR) [10].

In this present study, we examined chromatin changes in the human induced pluripotent stem cell (iPSC)-neural progenitor cell (NPC) differentiation model using Hi-C, a variation of the chromosome conformation capture (3C) method first published by Dekker et al. [9] and is distinguished from other adaptations such as 4C, 5C, ChIA-PET [16,17] by its unbiased genome-wide analysis of chromatin interaction frequencies [18]. The Hi-C library is generated by crosslinking chromatin, digesting crosslinked DNA using specific restriction enzymes, and the ligation of fragments; the resulting ligated fragments reflect actual proximal regions. After the sequencing and processing of the Hi-C library, one can obtain paired-end reads that contain information on physical interactions between pairs. Generally, intrachromosomal interactions are observed at a much higher rate compared to interchromosomal interactions, and interaction frequencies tend to diminish as two-dimensional genomic distance increases [19]. To date, there are scarce studies reporting on chromatin interaction in our iPSC-NPC model, making our findings a valuable resource in identifying gene networks regulated by spatial organization during lineage-specific differentiation from iPSCs.

Based on chromatin re-organization analysis, we have identified gene groups involved in neuronal differentiation, which are clustered non-randomly in networks that correlate with their biological function and expression level. Our results provide a resource to integrate chromatin interaction changes in understanding gene regulation during neuronal differentiation.

## 2. Methods and Materials

### 2.1. iPSC Cells Culture and Generation of iPSC-Derived Neural Progenitor Cells

The use of human iPSCs was approved by the Institutional Review Board (IRB) of Yonsei University (Permit Number: 7001988-201802-BR-119-01E). We cultured iPSC on Matrigel (354277, BD Biosciences (San Jose, CA, United States)) with Essential 8 medium (A1517001, Invitrogen (Carlsbad, CA, United States)) coating [20,21,22]. For the differentiation of iPSC into NPC, an embryoid body (EB) was generated through the culturing of human iPSCs for 5–6 days on non-adherent petri dishes in Essential 8 (Invitrogen) with additional supplement 5 μM dorsomorphin (DM) (P5499, Sigma-Aldrich (St. Louis, MO, United States)) and 5 μM SB431542 (Sigma-Aldrich)). Then we attached EBs in the new culture dish coated with Matrigel (BD Biosciences) with neural induction medium comprised of DMEM/F12 medium (11320033, Invitrogen), 1× N2 supplement (17502048, Invitrogen), 1× nonessential amino acids (11140050, Invitrogen) for 6 days. When neural rosette formation appeared in the center of the EBs, NPCs were collected for analysis [23,24,25].

### 2.2. iPSC and iPSC-Derived Neural Progenitor Cells Immunofluorescence Staining

The cells are permeabilized with 0.1% (*v/v*) Triton X-100 (X100, Sigma-Aldrich) in PBS for 20 min. Then, the cells were treated with 4% (*w/v*) bovine serum albumin (216006980, MP Biomedicals (Santa Ana, CA, United States)) for 10 min to block the non-specific binding of antibodies. The samples were incubated with the following primary antibodies overnight at 4 °C: rabbit anti-OCT-4 (1:500, AB3209, Millipore (Burlington, MA, USA)), mouse anti-Tra1-60 (1:500, MAB4360, Millipore), mouse monoclonal anti-Nestin (MAB5326, 1:1000; Millipore), mouse monoclonal anti-Paired box 6 (PAX6, 1:1000; DSHB (Iowa City, IA, USA)) and rabbit polyclonal anti-Ki67 (ab15580, 1:1000; Abcam). The stained samples were then washed with PBS three times, and incubated with secondary antibodies Alexa Fluor 488 or Alexa Fluor 594-conjugated secondary antibodies (1:200, A11001/A11012/A11005/A11008, Thermo Fisher Scientific (Waltham, MA, United States)) for 1 h in room temperature. Additionally, nuclei were counterstained with DAPI (A2412, TCI (Chuo-ku, Tokyo, Japan)) for 10 min in room temperature. The samples were mounted using a fluorescent mounting medium (H1400, Vector laboratories (Burlingame, CA, USA). Immunofluorescent images were examined under the Zeiss LSM880 and Zeiss LSM700 microscope (Zeiss LSM880/Zeiss LSM700, Carl Zeiss (Oberkochen, Germany)).

### 2.3. Quantitative PCR

Quantitative PCR (qPCR) reactions were conducted with triplicate for specific mRNA regions using KAPA SYBR FAST qPCR Master Mix (KK4600, KAPA biosystems (Wilmington, MA, United States)). The resulting signals were normalized for GAPDH primer set. The following real- time polymerase chain reaction (RT-qPCR) condition was used: 95.0 °C for 3 min (1 cycle), 94.0 °C for 10 s, 60 °C for 20 s and 72 °C for 30 s (40 cycles). The following primer sequences were used: 5′-GTGGAGGAAGCTGACAACAA-3′ and 5′-ATTCTCCAGGTTGCCTCTCA-3′ for Oct4 mRNA; 5′-ATAACCTTGGCTGCCGTGTC-3′ and 5′-AGCCTCCCAATCCCAAACAA-3′ for Nanog mRNA; 5′-AGTGAATCAGCTCGGTGGTGTCTT-3′ and 5′-TGCAGAATTCGGGAAATGTCGCAC-3′ for Pax6 mRNA; 5′-CAGCGTTGGAACAGAGGTTGG-3′ and 5′-TGGCACAGGTGTCTCAAGGGTA-3′ for Nestin mRNA; 5′-CCAGCAAGAGCACAAGAGGAAGAG -3′ and 5′-AGGAGGGGAGATTCAGTGTGGTG -3′ for GAPDH mRNA.

### 2.4. Hi-C Data Processing

Raw data of Hi-C were obtained from the Gene Expression Omnibus (GEO) database in GSM4798671, GSM4798672, GSM4798673 and GSM4798674. Hi-C data processing was executed by the described method [26]. Each single-ended reads of Hi-C data were mapped to the hg19 human reference genome using Burrows-Wheeler Alignment (BWA)-mem with default parameters and all mapped pair reads were merged to a paired-end aligned bam file using in-house script in order to process chimeric reads that span the ligation site [27]. We discarded PCR duplicates by Picard and low-quality reads (MAPQ < 10). To focus on cis-chromosomal interactions, we removed invalid Hi-C reads in accordance with criteria defined in earlier studies [10,28]. The process of removing experimental biases and normalizing interaction frequencies are described in the previous study [26]. The contact matrices at the scale of chromosomes and TADs were visualized by using Juicier software [29] to create .hic file from BAM files and FAN-C software [30] to generate contact matrix plots from .hic file.

### 2.5. Topologically Associated Domain Calling

Topologically associated domains (TADs) were obtained based on the directionality index (DI) algorithms by Dixon et al. [31]. We scanned the whole genome with 40 Kb binsize and 2 Mb window size to calculate DI, which is a metric for the biased distribution of interaction frequencies. Whole genome DIs were subjected to the hidden markov model (HMM) to obtain the bias states underlying DI distributions. The regions starting with consecutive downstream-biased states and ending with the consecutive upstream-biased states are inferred to be topologically associated domains.

### 2.6. Differentially Interacted Region Calling

In order to find the differentially interacting regions (DIRs), we utilized DiffHiC 1.9.8. [32]. We first counted repairs per pairs of genomic bins of 5kb length and filtered out low-abundance bin pairs, retaining bin pairs with an average log count-per-million (CPM) over 0. We then normalized the sample-specific biases, which are manifested as overall correlation between fold-change and average log CPM, by applying the offsets obtained from loess regression on MA plot. The biological variations between iPSC and NPC were modeled with generalized linear models based on the Quasi-Likelihood method, and the QL F-test was performed to test the significance of the differential interactions between conditions. The resulting *p*-values were corrected for multiple testing by the Benjamini–Hochberg method [33]. The bin pairs with false-discovery rate (FDR) < 0.05 were selected as DIRs and assigned to up-DIR or down-DIR categories depending on the direction of changes.

### 2.7. mRNA-Sequencing Data Processing

We utilized RNA-seq data of iPSC and NPC in Gene Expression Omnibus (GEO) database (GSE156723). Raw reads of all data were aligned to the human reference (hg19) using Tophat2 (v2.1.0). Then, we calculated the FPKM (Fragment Per Kilobase of transcript per Million mapped reads) using RSEM (RNA-Seq by Expectation Maximization, v1.2.31) at the gene level. We executed a normalization process using cuffnorm (v2.2.1) to reduce any technical bias among the samples.

## 3. Results

### 3.1. Interaction Patterns upon NPC Differentiation from iPSC on Genomic and Domain Levels

Before generating chromatin interaction data in NPC differentiation, we inspected our differentiation system from iPSC to NPC by confirming specific markers of each cell type. In previous studies, we already constructed iPSC maintenance and differentiation technique [23,24,34]. We re-confirmed the pluripotency of iPSCs using markers OCT4, E-cadherin (E-cad) and Tra-1-60 and verified NPC status through the elevated expression of Pax6, Sox2 and Nestin, with Ki67 as a cell-proliferation marker in NPC (Figure 1A). We also checked the mRNA level of pluripotency markers and NPC markers by qPCR assay in iPSC and NPC states (Figure 1B). Based on this system, we performed in situ Hi-C using two biological replicates of iPSC and its differentiated counterpart NPC, with pre-processing carried out as described under Methods. Our generated data are highly reproducible and robust, as indicated by the strong correlation (Pearson correlation coefficient 0.71 for iPSC replicates and Pearson correlation coefficient 0.63 for NPC replicates), consistent cis/trans ratio between replicates and the high number of uniquely mapped reads (Figure 1C,D, Appendix A). We found that most of the uniquely mapped read pairs were mapped in self (40%–60%), with the higher proportion of remaining reads being mapped in cis (20%–30%), followed by in trans (10%–20%) (Figure 1D). Reads mapped in trans were also higher in proportion in NPCs relative to iPSC. Finally, replicates were merged to achieve a higher resolution, and interaction frequencies on the genome level were visualized through the contact matrix at a binning size of 500 kb. While higher level compartment structures were maintained through differentiation, there were notable increases in long-range intrachromosomal interactions between the distant genomic regions upon differentiation from iPSC to NPC (Figure 1E,F, Appendix A).

### 3.2. Analysis of Topologically Associated Domains

For a scaled-down examination of local topology domains, we performed topologically associated domains (TADs) calling based on the DI-based method [31]. We obtained 3051 and 2611 TADs for iPSC and NPC, respectively. Most of the iPSC-TADs (*n* = 3036, 99.5%) overlapped with NPC-TADs (*n* = 2608, 99.8%) and vice versa (Figure 2A), as exemplified in the four consecutively conserved TADs within chromosome 2 (region: 141.72 Mbp~148.68 Mb) (Figure 2B). To further examine the overlap status, we searched the maximally overlapping TAD for each individual TAD, and found that 2510 TADs in iPSC and NPC showed maximum reciprocal overlap, that is, there were 2510 TAD pairs pointing towards each other as the maximally overlapping TAD. The overlap fractions of these TAD pairs were enriched at around 1.0 for both TADs in pair (Figure 2C). These together suggest that though basic layouts of 3D genomic topology in the form of TADs remain consistent, there exists an extensive re-organization process of overall chromatin structures to establish distant interactions.

### 3.3. Differentially Interacted Regions during NPC Differentiation

We subsequently searched for differentially interacting regions (DIRs) between iPSCs and NPCs using the DiffHiC package (Figure 3A, Appendix A). A total of 802 genomic regions exhibited significant enrichment in interaction (Up-DIRs), with 591 regions showing decreased interactions (Down-DIRs). Further analysis revealed that 1324 genes were associated with Up-DIRs, which were mostly enriched in neuronal differentiation-specific gene ontology terms (neuronal differentiation, brain development. etc.), whilst 1216 genes associated with Down-DIRs were related to gene ontology terms such as stem cell differentiation and embryonic patterning (Figure 3B). Such patterns of DIR association imply that re-organized spatial organization on the domain level plays a significant role in determining the direction of cellular identity program.

### 3.4. Role of Promoter–Promoter Interactome in Cell Fate Specification

Genes involved in the same cellular or biological functions are often clustered together in 3D forming separate chromatin domains, sometimes in phase-separated domains, for the fast and coherent regulation of their expression. We reasoned that these regulatory schemes can be explored by examining how promoters of genes are clustered together and thus comprehensively analyzed the correlation between promoter–promoter interactomes and gene expression in iPSC and NPC. We found that there was a significant enrichment for coherent status (both promoter pairs suppressed or both promoter pairs activated) and categorized coherently regulated promoter pairs into Group 1 (suppressed—both genes showed low expression, FPKM < 1) and into Group 2 (activated—both paired genes showed high expression, FPKM > 2) (Figure 4A). Gene ontology analysis of the 6255 interacting genes found in iPSC Group 1 (iPSC G1) and 9130 interacting genes in iPSC Group 2 (iPSC G2) showed association with differentiation and stem cell maintenance program, respectively (Appendix A). On the other hand, promoter pairs were related with lineage-specific/restricted expression in NPC groups (Group 1: non-ectodermal lineage differentiation, Group 2: ectodermal differentiation) (Appendix A). Given the above patterns, it is suggestive that while spatial organization affects gene expression, transcriptional machinery itself could reciprocate in affecting genome organization.

We also noticed that most of the iPSC G1 and iPSC G2 promoter pairs retained status quo even after NPC differentiation (G1: 5379/6255 and G2: 7908/9130). There was also a large set of promoters (*n* = 4985) that came to exhibit coherent regulation upon NPC differentiation, consistent with the general increase in the distal chromatin interactions from iPSC to NPC (Figure 4B). The increase in the expression level for genes associated with neuronal differentiation (iPSC G1 to NPC G2) is noteworthy (Figure 4C) as compared to genes less involved in such differentiation (iPSC G1 to NPC G1), as it implies that our classification is biologically meaningful.

## 4. Discussion

iPSCs are highly valued for their versatility in generating different cell types without the ethical concerns of human embryonic stem cells since adult somatic cells are used. This flexibility is appreciated to a great extent in the study of neurodegenerative disorders as diseased cells are challenging to obtain from patients. NPCs, namely precursor cells derived from iPSCs, also display great plasticity, which explains why they are the common platform from which various neurons are differentiated for utilization in disease modelling and treatment of a wide range of disorders such as Parkinson’s disease, Alzheimer’s disease, schizophrenia and spinal injury. Much light has been shed on this early differentiation process in the form of cellular, physiological and transcriptomics studies, which have described defining parameters and changing gene expression during differentiation. However, there are limitations in using end-point characterization in the further optimization of differentiation protocol, and analysis of non-coding regions is challenging using transcriptomic data since gene regulation mediated by such regions is independent of their linear position on the genome. Reviewing genome architecture on the third dimensional level would be ideal, and thus we observed changes in global chromatin organization using Hi-C, a derivative of 3C coupled with high-throughput sequencing. Since each read-pair reflects actual interaction between two proximal DNA regions, we were able to find biologically relevant gene sets involved in neuronal differentiation. Genome spatial organization is known to affect gene expression by clustering distal regulatory elements together, but, to the best of our knowledge, there is little published data on delineating genomic landscape changes during differentiation from iPSCs to NPCs. High-quality data using two biological samples of each cell type were generated, and we could detect a general increase in intrachromosomal interaction across all chromosomes during neuronal differentiation.

While genomic compartments are useful for understanding general organization principles, many biological processes take place at a smaller scale on the domain level. Topologically associated domains (TADs), which are functional units of chromosomes, form a framework within which promoters find their enhancers and vice versa [31]. We noted that while the basic TAD boundaries remained the same during neuronal differentiation, changes in distant interactions were observed. We thus examined to see if such interactions were relevant by calling differentially interacting regions (DIRs) as described under the methods section. DIRs confer greater meaning compared to just statistically significant interactions themselves as they are directly associated with biological condition and also less prone to being influenced by technical errors. The genes found in our up-regulated DIRs showed strong enrichment in neuronal differentiation gene ontology terms, indicating that spatial genome organization indeed does play a role in determining cellular identity direction.

We then reasoned that investigating promoter interactions would be logical since it is known that genes involved in the same cellular functions are often clustered together [11], and while much research has been conducted to identify long-range chromatin interaction between cis-regulatory elements, little is known about the gene candidates that are involved in neuronal differentiation. By examining the co-relationship between promoter–promoter interaction and expression level, we were able to able to observe four clusters—iPSC Group 1 (iPSC G1), iPSC Group 2 (iPSC G2), NPC Group 1 (NPC G1) and NPC Group 2 (NPC G2). G1 (FPKM < 1) and G2 (FPKM > 1) are differentiated by the expression level of both gene pairs, and gene ontology analysis of the four clusters suggests that our method of classification gives rise to biologically significant candidates. We then overlapped the four clusters together and found that the majority of gene pairs maintained coherent status (no change in expression level status) after differentiation. We were most interested in examining the gene pairs shared between iPSC G1 and NPC G2, as their increase in expression level after differentiation suggested that they would most likely be involved in ectodermal differentiation. Gene ontology analysis of this shared set supported our rationale as genes involved in neuronal differentiation were highly enriched.

Gene regulation can be summarized largely into three layers—genomic (chromosome territories), epigenomic (DNA and histone modification, DNA methylation) and epitranscriptomic (RNA modification) [35]. We have shown that there are changes on the genomic level during the iPSC-NPC differentiation process, which led to the identification of a specific gene group purported to be involved in neuronal differentiation. Our method of using chromatin interaction changes could be a plausible alternative to finding biologically significant candidates to further examine, other than basing on differential gene expression and network analysis. Finally, our data are a useful resource for further integrative analysis to tie the multiple layers of gene regulation and to better understand the relationship between chromatin dynamics and other genomic and epigenomic features. However, since our data were generated from one type of iPSC line, there are some limitations in generalizing our observations to encompass all stem cell types. Given that reprogramming methods have an influence on cell chromatin landscape, further studies using another iPSC line or natural embryonic stem cells such as H1 and H9 would be needed.

## 5. Conclusions

In summary, we showed that there is potential in identifying gene sets involved in neuronal differentiation based on both chromatin spatial organization and gene expression level. Analysis of differentially interacting regions adds another dimension to the search for biologically significant genes since actual interacting DNA regions are detected. We have provided robust and reliable Hi-C data that could be used as a resource to further study the regulation of the iPSC-NPC differentiation process, which is a critical step in creating patient-specific cells for both therapeutic and disease-modelling purposes.

## Figures and Tables

**Figure 1 genes-11-01176-f001:**
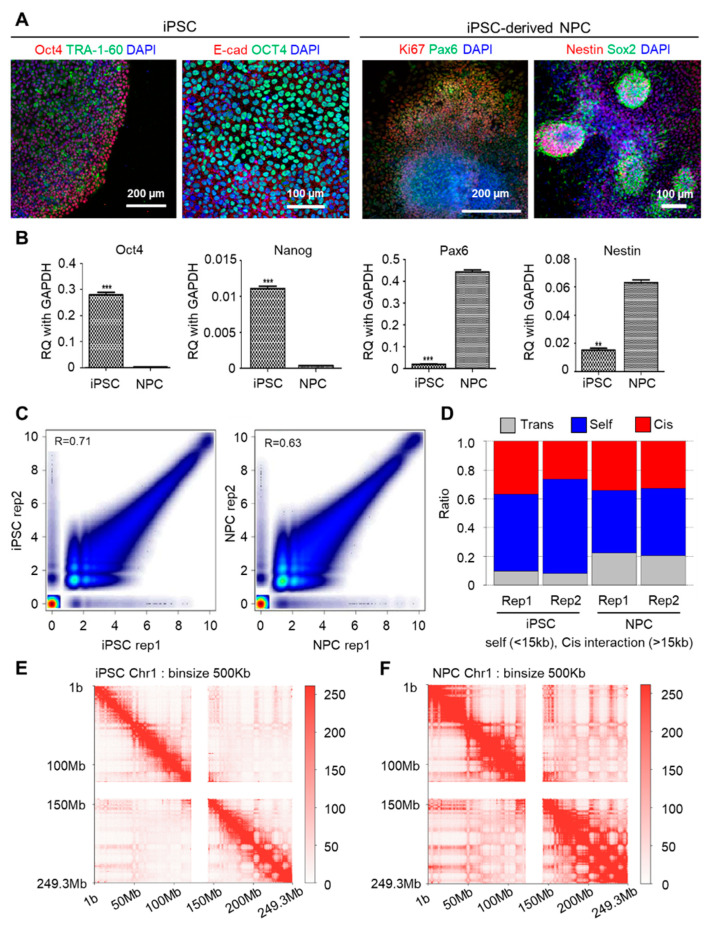
Genome-level interaction patterns observed using validated data. (**A**) Immunostaining of markers of iPSC (Tra-1-60, OCT4, E-Cadherin) and NPC (Nestin, Pax6, Ki67, Sox2). Scale bar is µm. (**B**) RT-qPCR markers for stem cell pluripotency markers and NPC markers. These data include the minimum 3 replicates. Relative quantification values were normalized by GAPDH. *** refers to *p*-value < 0.001 and ** refers to *p*-value < 0.01 (unpaired two tailed *t*-test). (**C**) Scatter plots showing the correlation of contact counts (Pearson correlation coefficient) between 2 biological replicates of iPSC and NPC, respectively. (**D**) Consistent cis/trans interaction ratio reflect accuracy of generated libraries and pre-processing. (**E**–**F**) Increase in intrachromosomal interactions during neuronal differentiation reflected in a contact matrix of a representative chromosome (chr1).

**Figure 2 genes-11-01176-f002:**
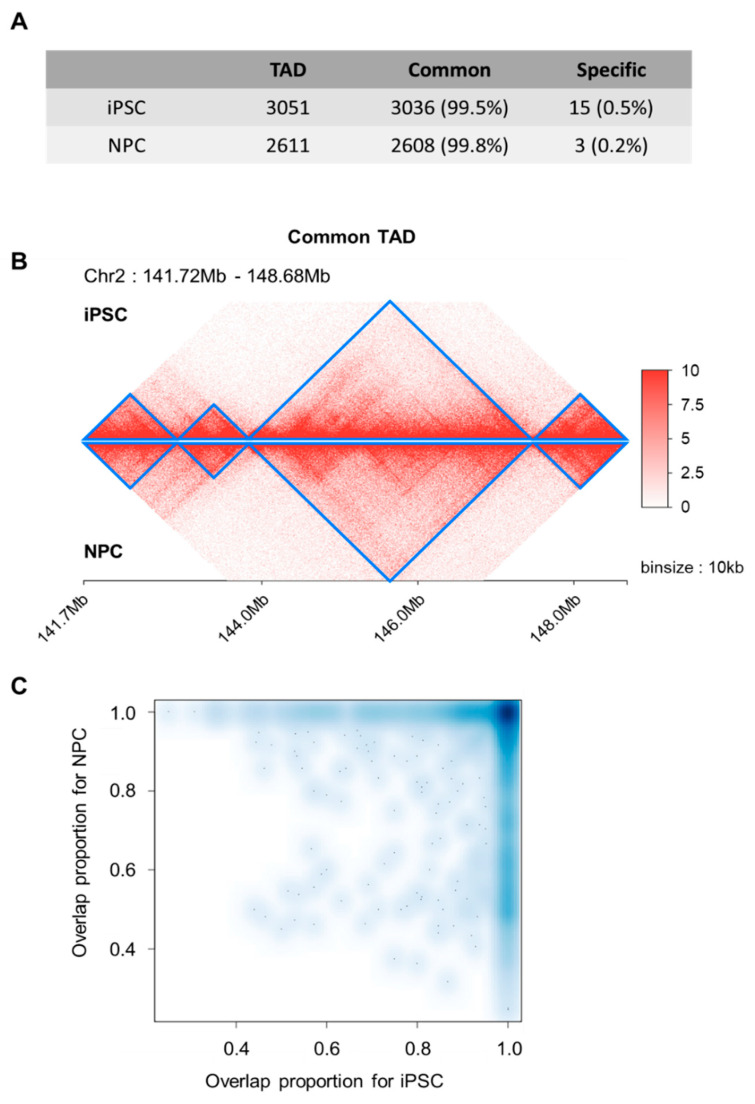
Domain-level chromatin interactions. (**A**) The vast majority of observed topologically associated domains (TADs) are shared between iPSC and NPCs. (**B**) Consecutively conserved TAD patterns in the 7 Mb region (marked in blue) during neuronal differentiation. (**C**) A reciprocal overlap fraction heatmap reflecting the overlap of TADs between iPSCs and NPCs.

**Figure 3 genes-11-01176-f003:**
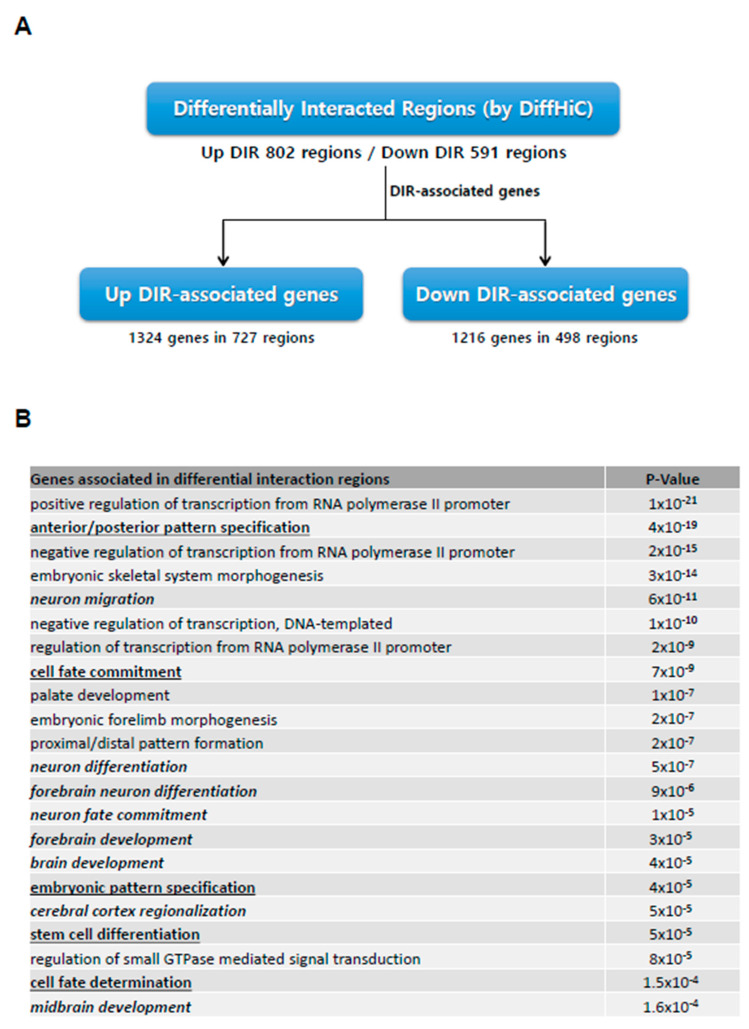
Significant differential interactions reveal role of spatial organization in iPSC to NPC differentiation. (**A**) The number of genes associated with differentially interacting regions (DIRs) derived using the DiffHiC package. (**B**) Gene ontology terms of associated genes from (**A**), whereby up-DIR associated genes are italicized while down-DIR associated genes have been underlined.

**Figure 4 genes-11-01176-f004:**
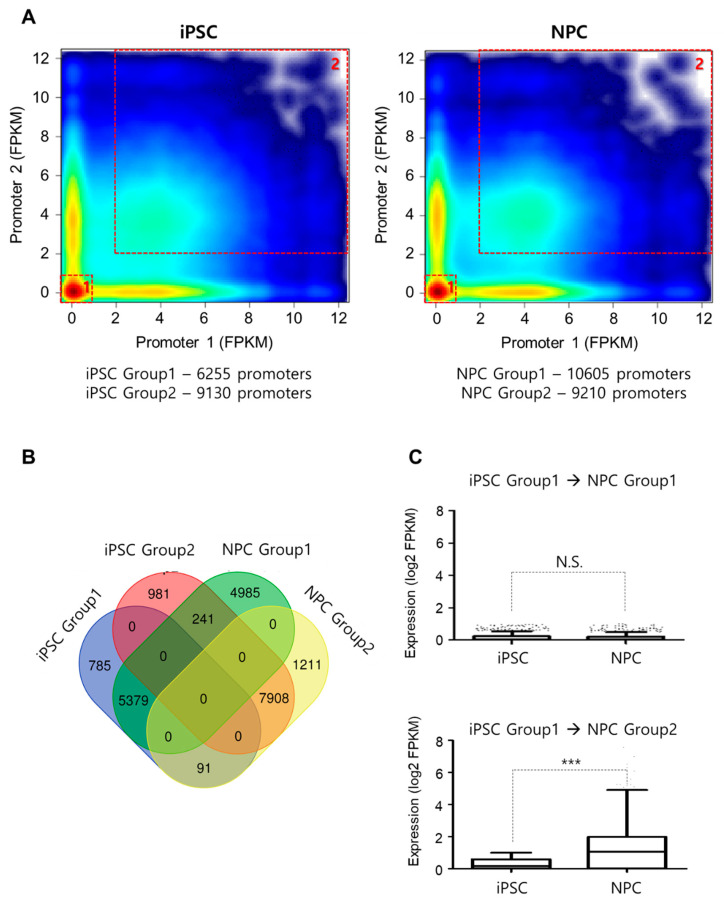
Role of promoter–promoter interactome in cellular identity program maintenance and development. (**A**) Correlation between promoter–promoter interactions and related gene expression shows formation of 2 distinct clusters in both iPSCs and NPCs, respectively—Group 1 (both promoter pairs suppressed, low FPKM) and Group 2 (both promoter pairs activated, high FPKM) enables the quick identification of current cell status. (**B**) The number of overlapping promoter–promoter interactions between groups. (**C**) Change in the expression level of genes (FPKM) shared between representative groups during differentiation. For all box-and-whisker plots, the horizontal lines in the boxes present the median value. The upper and lower error bars indicate the 90th and 10th percentiles, respectively. *** *p*-value < 0.001 (unpaired *t*-test, Mann–Whitney test, two-tailed).

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
