# Peer review of "Chromatin Interaction Changes during the iPSC-NPC Model to Facilitate the Study of Biologically Significant Genes Involved in Differentiation"

_genes, 2020, doi:10.3390/genes11101176_

Round 1

Reviewer 1 Report

The manuscript "Chromatin interaction changes during iPSC-NPC model to facilitate study of biologically significant genes involved in differentiation" by Won-Young Choi et al., evaluates the changes in chromatin organization occurring during human iPSCs differentiation into NCSs by using Hi-C coupled with high-throughput sequencing. As already stated by the authors, very little information is available on the mechanisms regulating chromatin changes during differentiation, while the Cellular and transcriptomic profiles have been widely speculated. The authors were able to identify biologically relevant gene sets playing a key role in neuronal differentiation. Interestingly, they offer an overview of the genomic landscape changes involved during hiPSCs differentiation to NSCs that can be exploit for further analysis of differentiation potential.

Thus, this work sounds very interesting to achieve a more comprehensive picture of the neuronal differentiation process.  Overall, the manuscript is quite well written and the data are well presented, but I still have some few comments:

The major point is that one single hiPSC line is to me not sufficient to draw robust conclusions. At least two different hiPSC lines should be used and I would recommend to include in the analysis also a line of hESCs (e.g. H1 or H9) as “natural” counterpart of induced pluripotent stem cells.

Minor considerations:

  1. Line 107: the word “model” is written twice within the sentence “DIs were subjected to hidden markov model (HMM) model
  2. The acronymous iPSC is sometimes written as hiPSC, iPSC, hiPSCs throughout the text. Please use the same acronymous to refer to human induced pluripotent stem cells (hiPSCs) all the time. The same consideration applies to NSCs as well

  1. Line 137: “Our generated data is highly….” , please change to “Our generated data ARE highly

  1. Line 292: the sentence “we were able to” is repeated twice. Please correct.

Reviewer 2 Report

In this manuscript authors examine chromatin changes using Hi-C in two cell types that are highly valued in the study of neurodegenerative diseases (hiPSCs and NPCs). Based on chromatin reorganization analyses, they have identified gene groups involved in neuronal differentiation, which are clustered non-randomly in networks that correlate their biological function and expression level. They showed that while topologically associated domains remained mostly the same during differentiation, presence of differential interacting regions in both cell types suggested that spatial organization affects gene regulation of both pluripotency maintenance and neuroectodermal differentiation. They provide a resource to integrate chromatin interaction changes in understanding gene regulation during neuronal differentiation.

Although manuscript could be of interest for the journal, an important issue must be addressed before accepting it.

Major changes:

  1. Before generating chromatin interaction data in NPC differentiation, it is critical to check the pluripotency of iPSCs and the potential of differentiation from iPSCs to NPCs. Authors claim that these important points have been confirmed by an immunofluorescence analysis (Figure 1). However, the pluripotency of iPSCs must be checked in a more reliable way. Note that pluripotency cannot be demonstrated with these images because they are not clear. Please, kindly provide other images acquired at a higher resolution and magnification and also include the pluripotency NANOG marker. The same applies to Ki67, PAX6 and DAPI. The proliferative NPCs are characterized by the expression of SRY (sex determining region Y)-box 2 (SOX2) and PAX6. Please, provide new images showing a higher resolution and magnification and an immunofluorescence showing the SOX2 marker.

Minor changes:

  1. Please include all the catalogue numbers of products in the material and methods section. This is important when anyone want to repeat experiments. Include also all the countries.
  2. Line 26, Please change “neurodegenerative disease” by neurodegenerative diseases”

Round 2

Reviewer 2 Report

Thank you for the new revised version of your manuscript genes-948281 entitled “Chromatin interaction changes during iPSC-NPC model to facilitate study of biologically significant genes involved in differentiation”. I am very grateful for your efforts to improve the manuscript content. However, the following modification is still needed:

Authors have added IF data of Nestin and Sox2 which are also crucial markers of NPC but again images are not well defined. Neural rosettes and also nuclei are diffused and not clear. Please, include a new image with higher resolution. In that way the quality of the manuscript clearly improves.
